# SHOT: Suppressing the Hessian along the Optimization Trajectory for Gradient-Based Meta-Learning

**JunHoo Lee, Jayeon Yoo, and Nojun Kwak**[*]
Seoul National University
`{mrjunoo, jayeon.yoo, nojunk}@snu.ac.kr`

## Abstract

In this paper, we hypothesize that gradient-based meta-learning (GBML) implicitly suppresses the Hessian along the optimization trajectory in the inner loop. Based on this hypothesis, we introduce an algorithm called SHOT (Suppressing the Hessian along the Optimization Trajectory) that minimizes the distance between the parameters of the target and reference models to suppress the Hessian in the inner loop. Despite dealing with high-order terms, SHOT does not increase the computational complexity of the baseline model much. It is agnostic to both the algorithm and architecture used in GBML, making it highly versatile and applicable to any GBML baseline. To validate the effectiveness of SHOT, we conduct empirical tests on standard few-shot learning tasks and qualitatively analyze its dynamics. We confirm our hypothesis empirically and demonstrate that SHOT outperforms the corresponding baseline. Code is available at: https://github.com/JunHoo-Lee/SHOT

## 1 Introduction

With the advent of *deep learning revolution* in the 2010s, machine learning has widened its application to many areas and outperformed humans in some areas. Nevertheless, machines are still far behind humans in the ability to learn by themselves, in that humans can learn concepts of new data with only a few samples, while machines cannot. Meta-Learning deals with this problem of *learning to learn.* An approach to address this issue is the metric-based methods Chen and He [2021], Snell et al. [2017], Bromley et al. [1993], Grill et al. [2020], Chen et al. [2020], Sung et al. [2018], Radford et al. [2021], which aims to learn 'good' kernels in order to project given data into a well-defined feature space. Another popular line of research is optimization-based methods Finn et al. [2017], Ravi and Larochelle [2017], Rusu et al. [2018a], so-called Gradient-Based Meta-Learning (GBML), which aims to achieve the goal of meta-learning by gradient descent.

As the most representative GBML methods applicable to any model trained with the gradient descent process, model-agnostic meta-learning (MAML) Finn et al. [2017] and its variations Raghu et al. [2019], Oh et al. [2020], Rusu et al. [2018b] exploit nested loops to find a good meta-initialization point from which new tasks can be quickly optimized. To do so, it divides the problem into two loops: the inner loop (task-specific loop) and the outer loop (meta loop). The former tests the meta-learning property of fast learning, and the latter moves the meta-initialization point by observing the dynamics of the inner loop. The general training is performed by alternating the two loops. In this paper, by focusing on the inner loop of GBML methods, we propose a new loss term for the outer loop optimization.

To properly evaluate the meta-learning property in the inner loop, GBML samples a new problem for each inner loop. Fig. 1 shows how the problem is formulated. For every inner loop, the task is

---

[*]Corresponding author.

37th Conference on Neural Information Processing Systems (NeurIPS 2023).

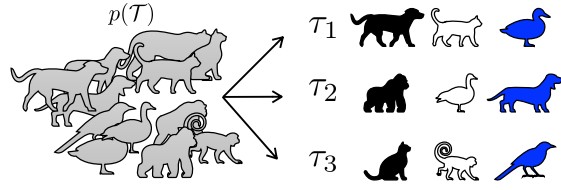

Figure 1: For the $i$-th inner loop in meta-learning, the task, $\tau_i$, is sampled from the task distribution: $N$ (e.g. 3) classes constituting each task are randomly sampled with a random configuration order, meaning that the model does not have any information before it retrieves an unseen task.

sampled from the task distribution. The classes constituting each task are randomly sampled, and the configuration order (class order) is also randomized. This means that the model tackles an entirely new task at the beginning of each inner loop. Since the task is unknown and the model has to solve the problem within a few gradient steps (some extreme algorithms Raghu et al. [2019], Oh et al. [2020] solve the problem in one gradient step), the model typically exploits large learning rates to adapt quickly. For example, MAML Finn et al. [2017] and ANIL Raghu et al. [2019] use significantly larger learning rates of 0.4 and 0.5 respectively, for the inner loop compared to that of the outer loop ($10^{-3}$). These rates are much larger than those used in modern deep learning techniques such as ViT Dosovitskiy et al. [2020], which typically use rates in the order of $10^{-6}$. However, using such large learning rates can lead to instability due to high-order terms like the Hessian LeCun et al. [2015]. The issue of instability caused by high-order terms is particularly evident during the initial training phase. To mitigate this problem, some deep-learning methodologies employ a heuristic that utilizes a smaller learning rate at the beginning Dosovitskiy et al. [2020], He et al. [2018]. Given that GBML goes through only several optimizations in the entire inner loop, GBML's inner loop corresponds to the initial training phase. Therefore, we anticipate that the problem caused by high-order terms may be severe with a large learning rate in each inner loop. However, it is noteworthy that GBML appears to perform well despite these potential issues.

In this paper, we investigate the inner loop of GBML and its relationship with the implicit risk of using a large learning rate with a few optimization steps in the inner loop. We observe that the gradient of a GBML model remains relatively constant within an inner loop, indicating that the Hessian along the optimization trajectory has a minimal impact on the model's dynamics as learning progresses. We hypothesize that this phenomenon is a key property that enables GBML to mitigate the risk of using a large learning rate in the inner loop.

Our hypothesis that GBML implicitly suppresses the Hessian along the optimization trajectory suggests a desirable property for a GBML algorithm. By explicitly enforcing this property instead of relying on implicit suppression, we anticipate that the model will achieve faster convergence and superior performance. However, enforcing this property is not trivial as it involves dealing with high-order terms, which naturally require a significant amount of computation. In this paper, we propose an algorithm called *Suppressing the Hessian along the Optimization Trajectory* (SHOT) that suppresses the Hessian in the inner loop by minimizing the distance of the target model from a reference model which is less influenced by the Hessian. Our algorithm can enforce the Hessian to have desirable properties along the inner loop without much increasing the computational complexity of a baseline model. More specifically, SHOT only requires one additional forward pass without any additional backward pass in general cases.

Our algorithm, SHOT, is agnostic to both the algorithm and architecture used in GBML, making it highly versatile and applicable to any GBML baseline. To validate the effectiveness of SHOT, we conducted empirical tests on standard few-shot learning tasks, including miniImagenet, tiered-Imagenet, Cars, and CUB Vinyals et al. [2016], Ren et al. [2018], Krause et al. [2013], Welinder et al. [2010]. Furthermore, we tested SHOT's cross-domain ability by evaluating its performance on different benchmarks. Also, SHOT can be applied to one-step or Hessian-free baselines, acting as a regularizer. Our results demonstrate that SHOT can significantly boost GBML in an algorithm- and architecture-independent manner. Finally, we analyzed the dynamics of SHOT and confirmed that it behaves as expected.

## 2   Related Works

GBML, or optimization-based meta-learning, is a type of methodology which wants to mimic people's rapid learning skill. Thus, a model should learn with a small number of samples and gradient

steps. To achieve this property, Ravi and Larochelle [2017] tried to save meta-data to recurrent networks. Recently, MAML-based methods Finn et al. [2017], Rajeswaran et al. [2019], Rusu et al. [2018b], Baik et al. [2020], Bernacchia [2021] are typically considered as a synonym of GBML consisting of nested loops: outer loop (meta-loop) and inner loop (task-specific loop). They sample few-shot classification tasks to test the meta-learning property in the inner loop and update the meta-initialization parameters in the outer loop with SGD. To learn rapidly, they exploit a more-than-thousand times larger learning rate in the inner loop compared to modern deep learning settings such as ViT Dosovitskiy et al. [2020] and CLIP Radford et al. [2021]. Although some researches have proved their convergence property Fallah et al. [2020], Wang et al. [2020], they set infinitesimal size of learning rate to prove convergence. Fallah et al. [2020] set the learning rate considering the Hessian *i.e.*, setting the learning rate small enough so that the higher-order terms do not affect the convergence and Wang et al. [2020] proved the convergence with the assumption that the learning rate goes to zero. Although some lines of research have shown algorithms dealing with learning rates Baik et al. [2020], Bernacchia [2021], they did not explain the success of a large learning rate. Baik et al. [2020] used an additional model to predict a proper task-specific learning rate. ANIL (Almost No Inner Loop) Raghu et al. [2019] and BOIL (Body Only Inner Loop) Oh et al. [2020] tried to explain GBML with the dynamics of the encoder's output features. Although both used the same feature-level perspective, they argued in exactly different ways. ANIL Raghu et al. [2019] argued feature reuse is an essential component, while BOIL Oh et al. [2020] said that feature adaption is crucial. Both algorithms can be considered as a variant of preconditioning meta-learning since they freeze most layers in the inner loop, thereby frozen layers can be considered as warp-parameters Flennerhag et al. [2019]. Our SHOT can be viewed as a variant of preconditioning, as we hypothesize that there exists a suitable condition of meta-parameter that can adapt well to a given meta-learning task. However, unlike previous work, we define the condition of the meta-parameter based on the curvature of the loss surface. Additionally, our algorithm does not require any specific architecture such as warp-parameters, making it applicable to any architecture and any GBML algorithm. Finally, there exists some paper which delt with curvature, those all required some special architecture Simon et al. [2020], or explicit calculation of the Hessian Hiller et al. [2022], Park and Oliva [2019]. In contrast, we propose much weaker condition and does not require specific architecture. We only suppress the Hessian along the inner loop. In addition, we discuss in the Appendix the possibility of integrating the hypotheses proposed in ANIL Raghu et al. [2019] and BOIL Oh et al. [2020] into our SHOT framework.

Optimization methods based on second-order or first-order derivatives that take curvature into account is a vast area of research. Although they incur a higher computational cost than vanilla SGD, they have been demonstrated to exhibit superior convergence properties. Several studies have investigated the curvature of the loss surface while maintaining the first-order computational costs. Natural gradient descent Bonnabel [2013], a type of Riemannian gradient descent employing the Fisher information metric as a measure, is coordinate-invariant, and exhibits better convergence properties than vanilla gradient descent. Given that GBML utilizes relatively fewer samples, incorporating second-order terms is computationally more feasible. As a result, there is a growing body of research that employs natural gradient descent in the inner loop. Additionally, the SAM optimizer Foret et al. [2020] implicitly leverages curvature by iteratively searching for flat minima during training, leading to better generalization properties than SGD. This means they find a point where the Hessian is close to zero around a local minimum. Our method can be interpreted as the variation of Knowledge Distillation Hinton et al. [2015]. We reduced the distance between models, slow model and fast model. However, we do not need any pretrained model. Those models share same meta-initialization parameter.

As our goal is to find a proper meta-initialization point that is not affected by the Hessian along the inner loop, our proposed algorithm, SHOT, also deals with the Hessian of the loss surface. However, unlike SAM optimizer Foret et al. [2020], we do not seek a point where the Hessian is zero in any direction. We rather seek a point where the Hessian does not take effect only along the optimization trajectory in the inner loop. By using this restriction, we give a looser constraint. Also, while the SAM optimizer has a nested loop to find a proper point, SHOT does not need any iteration in a single optimization step. Also, although natural gradient descent regards curvature like our algorithm, our SHOT does not calculate the exact Hessian during inference time as we precondition the Hessian and only consider the Hessian's effect along the optimization trajectory. In this paper, we aim to shed light on how GBML's inner loop works. Through an analysis of the curvature of the loss surface in the inner loop, we hypothesize that the primary role of the outer loop is to implicitly regulate the

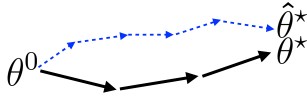

Figure 2: This figure provides the concept of the proposed SHOT. Specifically, a reference model ($\hat{\theta}^\star$) is initialized with the same support set and starting point as our model ($\theta^\star$), but undergoes significantly more optimization steps. The outer loop then minimizes the distance between the two models.

curvature of the inner loop's loss surface. To reinforce this property by explicitly incorporating a corresponding loss term in the inner loop, we propose an algorithm called SHOT, which takes into account the curvature of the loss surface in the inner loop and aims to enhance its generalization properties and dynamics by controlling the curvature itself. We provide empirical evidence supporting our hypothesis through the success of SHOT. Additionally, we offer a new interpretation of GBML as a variant of MBML in the Appendix.

## 3 Preliminaries: GBML

We consider the few-shot classification task of $N$-way $K$-shot, which requires learning $K$ samples per class to distinguish $N$ classes in each inner loop. In few-shot GBML, we can apply gradient descent for only a limited number of steps for fast adaptation.

Let $f(x|\theta) \in \mathbb{R}^N$ be the output of the classifier parameterized by $\theta$ for a given input $x$. In this setting, $f(\cdot)$ can be considered as the logit before the softmax operation. The model parameters $\theta$ are updated by the loss, $L(x, y|\theta) = D(s(f(x|\theta)), y)$, where $x$ and $y$ are the input and the corresponding label, $s(\cdot)$ is the softmax operation and $D(\cdot, \cdot)$ is some distance metric.

As shown in Fig. 1, at each inner loop, we sample a task $\tau \sim p(\mathcal{T})$, an $N$-way classification problem consisting of two sets of labeled data: $\mathcal{X}_s^\tau$ (support set) and $\mathcal{X}_t^\tau$ (target set). In the inner loop, the goal is to find the task-specific parameter $\theta_\tau^\star$ which minimizes the loss $L$ for the support set $\mathcal{X}_s^\tau$ given the initial parameter $\theta_0$ as follows:

$$\text{Inner Loop:} \quad \theta_\tau^\star = \arg\min_\theta \sum_{(x,y) \in \mathcal{X}_s^\tau} L(x, y|\theta; \theta_0). \tag{1}$$

Normally, $\theta_\tau^\star$ is obtained by SGD [Schmidhuber, 1987, Rusu et al., 2018a, Oh et al., 2020, Raghu et al., 2019]: $\theta^{k+1} = \theta^k - \alpha \nabla_\theta L(\theta^k)$, where $k$ indicates an inner loop optimization step and $\alpha$ is a learning rate. We use $T$ steps in the inner loop.

Then, for the outer loop, we optimize the initialization parameters $\theta_0$ which improve the performance of $\theta_\tau^\star$ on $\mathcal{X}_t$ as follows:

$$\text{Outer Loop:} \quad \theta_0^\star = \arg\min_{\theta_0} \sum_\tau \sum_{(x,y) \in \mathcal{X}_t^\tau} L(x, y|\theta_\tau^\star; \theta_0). \tag{2}$$

Although GBML gained its success with this setting, this success is quite curious: since $N$ classes constituting each task are randomly sampled and the configuration (class) order is also random, 'What characteristics of meta-initialization $\theta_0$ make GBML work?' remains quite uncertain.

## 4 Optimization in GBML

### 4.1 Hypothesis: Outer loop implicitly forces inner loop's loss surface linear

The perspective of gradient flow provides a way to understand the dynamics of optimization. In this view, the evolution of model parameters is seen as a discretization of continuous time evolution. Specifically, the equation governing the model parameters is $\dot{\theta} = -\nabla_\theta L(\theta)$, where $\theta$ represents the model parameter and $L(\theta)$ is the loss function. With gradient flow, we can characterize our models into two principal types: the **Fast Model** and the **Slow Model**. These definitions emerge from our observation of the models' convergence timelines. Considering the terminal point in time as 1 and the initialization as 0, we deduce the timestamp of a singular learning step as $\frac{1}{N}$, where N signifies the number of steps.

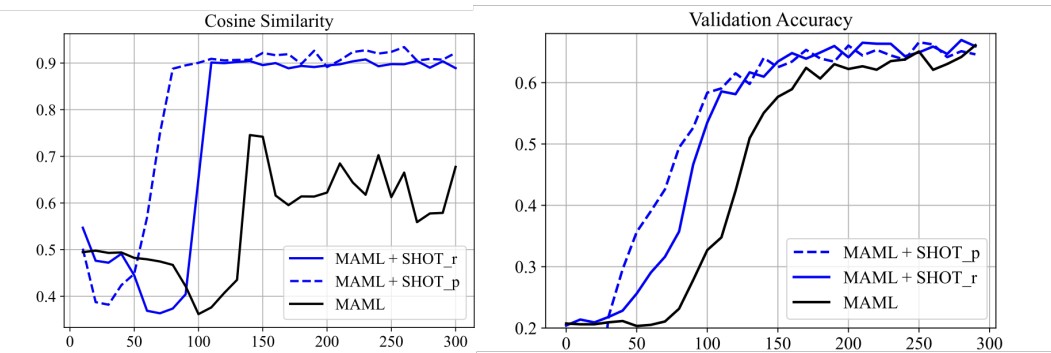

Figure 3: (left) **Averaged cosine similarity** between the overall parameter difference ($\theta_\tau^\star - \theta_0$) and each step's gradient ($\nabla_{\theta^k} L(\theta^k)$) in the inner loop over the training (meta) epochs. The cosine similarity increases as training progresses, indicating that the gradient does not vary much during an inner loop. (right) **Validation accuracy vs. training epoch**. To ensure a fair comparison, we shifted $SHOT_p$ for 30 epochs as it pretrains for 30 epochs with $L_{SHOT}$ only. For this figure, we trained $SHOT_r$ for 300 epochs to compare it at a glance. Our results show that SHOT converges much faster than the baseline. Additionally, we observed a strong correlation between cosine similarity and validation accuracy.

**Fast Model**: A model with a relatively small value of N is should converge rapidly. A salient exemplar is GBML's inner loop, typically evolving within a minuscule span of 1 to 5 optimization steps. Such rapid convergence necessitates an elevated learning rate.

**Slow Model**: In contrast, models with a larger value of N take longer to converge. Traditional deep learning models fall into this category. They require many gradient steps and a lower learning rate.

This separation makes clear the differences between typical deep learning models and GBML's inner loop. However, we should note that GBML's high learning rate approach can arise some issues.

$$\int_0^1 \nabla L(\theta(t)) \cdot \nabla L(\theta^k) dt \approx \int_0^1 \|\nabla L(\theta^k)\|_2^2 - \alpha t \nabla L(\theta^k)^T H(\theta^k) \nabla L(\theta^k) dt > 0, \qquad (3)$$

Gradient descent is the notion of treating the model as a (locally) linear approximation, adjusting its parameters in a direction where the loss decreases most rapidly. If a model adheres to Eq. 3, we can expect the gradient descent effective in that context. (Refer to Appendix A for the detailed proof.) However, the inner loop of GBML can be characterized as a fast model. This implies a pronounced influence of the Hessian in the optimization dynamics. Due to this, we cannot guarantee loss decreases in the inner loop. In this paper, We argue that the outer loop's primary role in GBML is to counteract this influence of the Hessian. In more specific terms, it aims to attenuate the impact of $H(\theta^k) \nabla L(\theta^k)$ on the optimization trajectory.

Fig. 3 supports our claim. As mentioned before, the optimization steps are a discrete representation of the gradient over time. This allows us to track how the gradient changes as time progresses. In meta-learning, the gradient's direction and magnitude do not change significantly as time progresses. This is because meta-learning algorithms are designed to learn how to quickly adapt to new tasks. By learning how to adapt to new tasks, meta-learning algorithms can avoid the need to learn from scratch on each new task. In contrast, in conventional deep learning, the gradient's direction and magnitude change significantly as time progresses. This is because conventional deep learning algorithms are not designed to learn how to quickly adapt to new tasks. Instead, conventional deep learning algorithms are designed to learn a general model that can perform well on a wide variety of tasks. The difference in how the gradient changes over time between meta-learning and conventional deep learning is one of the key factors that explains why meta-learning algorithms are able to learn new tasks so quickly.

It seems clear that Eq. 3 plays a crucial role in GBML, suggesting that we can enhance its performance and dynamics by explicitly inducing the model to meet this condition. However, the challenge remains: how can we directly reduce the Hessian?

Table 1: Comparison of the test accuracies (%) of a 4-convolutional network with random initialization and SHOT. SHOT does not receive label information from the outer loop. (*i.e.*, SHOT$_r$ without supervision ) The numbers in parentheses indicate the number of shots.

| meta-train | miniImageNet | | | Cars | | |
|---|---|---|---|---|---|---|
| meta-test | miniImageNet | tieredImageNet | Cars | Cars | CUB | miniImageNet |
| Random-init (1) | $21.36 \pm 0.01$ | $21.20 \pm 0.01$ | $21.10 \pm 0.93$ | $21.20 \pm 0.93$ | $21.62 \pm 0.02$ | $21.36 \pm 0.01$ |
| SHOT-init (1) | $\mathbf{24.62} \pm 0.04$ | $\mathbf{23.80} \pm 0.06$ | $\mathbf{24.63} \pm 0.04$ | $\mathbf{25.84} \pm 0.06$ | $\mathbf{25.86} \pm 0.03$ | $\mathbf{26.48} \pm 0.04$ |
| Random-init (5) | $21.29 \pm 0.09$ | $21.72 \pm 0.19$ | $21.58 \pm 0.37$ | $21.58 \pm 0.04$ | $22.31 \pm 0.06$ | $21.28 \pm 0.09$ |
| SHOT-init (5) | $\mathbf{27.32} \pm 0.59$ | $\mathbf{31.28} \pm 1.68$ | $\mathbf{29.10} \pm 1.81$ | $\mathbf{28.28} \pm 0.79$ | $\mathbf{30.34} \pm 2.69$ | $\mathbf{26.61} \pm 1.05$ |

## 4.2 SHOT (Suppressing Hessian along the Optimization Trajectory)

In this part, we present our novel algorithm, Suppressing Hessian along the Optimization Trajectory (SHOT), which is the main contribution of our paper. SHOT is designed to obviate the effect of the Hessian along the inner loop while maintaining the order of computation cost.

As previously discussed, directly reducing the Hessian is impractical. However, we can reduce the effect of the Hessian along the optimization trajectory, specifically by minimizing the product $H(\theta_t)\nabla L(\theta_t)$, which is in line with the condition expressed in Eq. 3 and computationally feasible. To achieve this, we define a measure that quantifies the effect of the Hessian along the optimization trajectory.

To do so, we first create a reference model that is not influenced by the Hessian. We obtain this reference model by increasing the number of optimization steps. The intuition behind this is that as we increase the number of optimization steps, the smaller the discretized step becomes, which means that the model with many optimization steps is less influenced by the Hessian. We then define the measure as the distance between the target model (a model with few optimization steps, $\theta_t$) and the reference model (a model with many optimization steps, $\theta_r$). This distance metric, denoted as $D(\theta_t, \theta_r)$, quantifies the distortion along the optimization path and evaluates the deviation from the ideal optimization trajectory. By minimizing this measure, we can achieve our goal of reducing the effect of the Hessian along the optimization trajectory. We will call this method as SHOT (Suppressing Hessian along the Optimization Trajectory) which adds the following loss term in the outer loop of a conventional GBML algorithm:

$$L_{SHOT} = D(\theta_t^T, \theta_r^R). \tag{4}$$

Here, $T$ and $R$ denote the number of steps in an inner loop used to obtain the target and reference model respectively. In this paper, we used KL divergence ($KL(s(f(\theta_t^T))|s(f(\theta_r^R)))$) as a distance metric $D$, but we also conducted experiments with other distance metrics such as $L_2(||\theta_t^T - \theta_r^R||_2^2)$ and cross-entropy ($-\sum_c s(f(\theta_t^T))_c \log s(f(\theta_r^R))_c$, as shown in Table. 6. We set $T < R$ because our intention is to measure distortion caused by the Hessian in the inner loop. As we increase the number of optimization steps in the inner loop, the effect of distortion by the Hessian decreases. Therefore, we could use SHOT as a pseudo-measure of distortion in the inner loop caused by the Hessian. To match the scale of distribution between $\theta_t^T$ and $\theta_r^R$, we set $\alpha$ in Eq. 1 as $\alpha_r = \frac{T}{R}\alpha_t$.

Reducing the effect of Hessian along the optimization in the inner loop, as measured by Eq. 4, can lead to faster convergence and better performance. By minimizing the deviation from the ideal optimization trajectory, SHOT can effectively enforce the condition expressed in Eq. 3, which is critical for the success of GBML. As shot is the measure of distortion in the whole trajectory of an inner loop. We use SHOT at the outer loop, where **scoring** of each optimization trajectory is done.

Table 1 highlights the importance of distortion in GBML. We compared the performance of a randomly-initialized model with that of a lower-distortion-initialized model, where we only reduced the distortion introduced in Eq. 4 in the outer loop. As our SHOT loss is self-supervisory that can be applied without target class labels, no target labels are given here. The results demonstrate that the lower-distortion-initialized model outperforms the randomly-initialized model, indicating that distortion is a crucial factor in GBML. This also suggests that we can generally improve GBML by incorporating our loss term, SHOT.

### 4.3 SHOT can be used as a regularizer or meta-meta-initializer in the outer loop.

One potential approach to incorporating SHOT is to use it as a regularizer in GBML. This approach involves adding SHOT to the loss function of the outer loop during training, thereby leveraging it as a regularizer reducing the effect of the Hessian along the optimization trajectory. Alternatively, SHOT can be utilized as a meta-meta-initializer as is used to obtain Table 1. In this approach, the entire model is pre-trained with SHOT, $i.e.$, The outer loss is substituted with $L_{SHOT}$ without utilizing any labeling information. Then the model is trained with the original loss function. However, there is a vulnerability to homogeneous corruption, where the loss goes to zero only if $\theta_t$ and $\theta_r$ produce the same output of all inputs. As this phenomenon is trivial in the self-supervised learning area, we drew inspiration from BYOL Grill et al. [2020]. To address this issue, we added a projector to the reference model, while the target remained the same. We also augmented $\mathcal{X}_t^\tau$. After pre-training with SHOT, we started training by loading the best model with the highest validation accuracy. To differentiate between SHOT utilized as a regularizer and SHOT utilized as a meta-meta-initializer, we denote them as $SHOT_r$ and $SHOT_p$, respectively.

### 4.4 SHOT does not require any computation burden

Several meta-learning algorithms have already assumed linearity in the inner loop. For example, Fo-MAML and Reptile Finn et al. [2017], Nichol et al. [2018] proposed Hessian-Free GBML algorithms that both assumed linearity in the inner loop. Additionally, ANIL Raghu et al. [2019] proposed a feature reuse concept that allows for translation to zero distortion in the inner loop. However, none of these studies explicitly argued that GBML implicitly suppresses the Hessian in the inner loop as an inductive prior, nor did they propose an algorithm that controls the Hessian. Our algorithm is similar to the 'SAM' optimizer Foret et al. [2020] or natural gradient descent Bonnabel [2013]. Like SAM and natural gradient descent, our algorithm searches for a good initialization point based on its curvature. However, SHOT finds it without much increasing the computational cost compared to normal gradient descent, which is a notable advantage over SAM or natural gradient descent. Although SAM uses a first-order computation term, it searches for flat minima by iteratively adding noise to each point. Natural gradient descent exploits the Fisher metric, which is a second-order computation cost.

On the other hand, we can make SHOT not give any computation burden. When we set the reference model $\theta_r$ to exploit the same number of optimization steps in the inner loop as the baseline model and the target model $\theta_t$ exploits one optimization step in the inner loop, because both models share the same initialization point $\theta^0$, $i.e.$, $\theta_t^0 = \theta_r^0$. $\theta_t$ can be obtained by calculating the first optimization step of $\theta_r$. Therefore, SHOT does not require any additional backpropagation. What we require is only one more inference at the endpoint $\theta_t^T$ for $SHOT_r$. Also, even if we set $\theta_r$ to have more optimization steps than the baseline model and set $\theta_t$ to have the same number of optimization steps as the baseline model, the computation cost at test time is unchanged as it is only used in the training phase.

## 5 Experiments

**Datasets** Our method is evaluated on miniImageNet [Vinyals et al., 2016], tieredImageNet Ren et al. [2018], Cars Krause et al. [2013], and CUB Welinder et al. [2010] datasets. miniImageNet and tieredImageNet are subsets of ImageNet Russakovsky et al. [2015] with 100 classes each and can be considered as general datasets. Cars and CUB are widely used benchmarks for fine-grained image classification tasks as they contain similar objects with subtle differences. By conducting experiments on these datasets, we can evaluate the performance of our method across a range of tasks.

**Architectures** We use 4-Conv from Finn et al. [2017] and ResNet-12 from Oreshkin et al. [2018] as our backbone architecture. We trained 4-Conv for 30k iterations and ResNet-12 for 10k iterations.

**Experimental Setup** To ensure a fair comparison, we fine-tuned MAML Finn et al. [2017] first, and our baselines outperformed the original paper. We used Adam Kingma and Ba [2014] as our optimizer and fixed the outer loop learning rate to $10^{-3}$ and inner loop learning rate to 0.5 for 4-Conv, and $10^{-5}$ and 0.01 for ResNet-12. We restricted the total number of steps in the inner loop to 3, as argued in Finn et al. [2017]. We set the number of steps to 1 for the target model $\theta_t$ and 3 for the reference model $\theta_r$ in our SHOT. With this setting, we can exploit SHOT without any additional computation burden as discussed in Sec 4.4. For $SHOT_r$, we fixed $\lambda$ to 0.1 when we set the original

Table 2: Test accuracy % of 4-conv network on benchmark data sets. The values in parentheses indicate the number of shots. The best accuracy among different methods is bold-faced.

| Domain | General (Coarse-grained) | | Specific (Fine-grained) | |
|---|---|---|---|---|
| Dataset | miniImageNet | tieredImageNet | Cars | Cub |
| MAML (1) | $47.88 \pm 0.55$ | $46.93 \pm 1.07$ | $47.78 \pm 0.99$ | $57.04 \pm 1.42$ |
| MAML + SHOT$_r$ (1) | $47.97 \pm 0.71$ | $47.53 \pm 0.59$ | $\mathbf{50.44} \pm 0.62$ | $57.55 \pm 0.64$ |
| MAML + SHOT$_p$ (1) | $\mathbf{48.11} \pm 0.26$ | $\mathbf{47.53} \pm 0.68$ | $49.08 \pm 0.88$ | $\mathbf{58.23} \pm 0.39$ |
| MAML (5) | $64.81 \pm 1.63$ | $66.12 \pm 1.10$ | $62.24 \pm 2.01$ | $72.48 \pm 0.86$ |
| MAML + SHOT$_r$ (5) | $\mathbf{66.86} \pm 0.58$ | $\mathbf{69.08} \pm 0.31$ | $64.20 \pm 1.58$ | $\mathbf{73.38} \pm 0.32$ |
| MAML + SHOT$_p$ (5) | $66.35 \pm 0.27$ | $68.94 \pm 0.87$ | $\mathbf{64.84} \pm 2.87$ | $73.27 \pm 0.40$ |

Table 3: Test accuracy % of 4-conv network on cross-domain adaptation. The values in parentheses indicate the number of shots. The best accuracy among different methods is bold-faced.

| adaptation | General to General | | General to Specific | | Specific to General | | Specific to Specific | |
|---|---|---|---|---|---|---|---|---|
| meta-train | tieredImageNet | miniImageNet | miniImageNet | miniImageNet | Cars | Cars | CUB | Cars |
| meta-test | miniImageNet | tieredImageNet | Cars | CUB | miniImageNet | tieredImageNet | Cars | CUB |
| MAML (1) | $47.52 \pm 1.66$ | $51.84 \pm 0.24$ | $34.41 \pm 0.47$ | $40.91 \pm 0.57$ | $28.67 \pm 1.17$ | $30.79 \pm 1.17$ | $32.74 \pm 1.12$ | $30.95 \pm 1.41$ |
| MAML + SHOT$_r$ (1) | $48.20 \pm 0.64$ | $51.68 \pm 0.68$ | $34.03 \pm 1.11$ | $41.58 \pm 0.56$ | $\mathbf{30.19} \pm 0.64$ | $\mathbf{31.96} \pm 0.50$ | $33.11 \pm 0.41$ | $31.21 \pm 0.64$ |
| MAML + SHOT$_p$ (1) | $\mathbf{48.45} \pm 0.13$ | $\mathbf{51.97} \pm 0.25$ | $\mathbf{34.84} \pm 0.16$ | $\mathbf{41.65} \pm 0.70$ | $29.02 \pm 0.73$ | $31.26 \pm 0.54$ | $\mathbf{33.69} \pm 0.45$ | $\mathbf{31.65} \pm 0.39$ |
| MAML (5) | $66.86 \pm 1.58$ | $67.96 \pm 1.22$ | $46.57 \pm 0.53$ | $56.32 \pm 1.17$ | $37.23 \pm 1.95$ | $41.00 \pm 1.86$ | $44.02 \pm 2.29$ | $41.84 \pm 1.25$ |
| MAML + SHOT$_r$ (5) | $\mathbf{70.70} \pm 0.29$ | $69.48 \pm 0.17$ | $\mathbf{48.42} \pm 0.72$ | $\mathbf{58.40} \pm 0.48$ | $\mathbf{40.79} \pm 0.93$ | $\mathbf{42.73} \pm 0.32$ | $\mathbf{44.47} \pm 0.25$ | $\mathbf{43.46} \pm 0.89$ |
| MAML + SHOT$_p$ (5) | $69.88 \pm 0.53$ | $\mathbf{69.83} \pm 0.34$ | $47.99 \pm 2.10$ | $58.30 \pm 0.44$ | $37.96 \pm 1.14$ | $41.94 \pm 0.23$ | $43.83 \pm 1.06$ | $40.64 \pm 1.25$ |

model as $\theta_r$. For one-step algorithms, we set the number of optimization steps to 2 for $\theta_r$ and 1 for $\theta_t$. So actual optimization step in inference time is 1, which is the same as the baseline. We set $\lambda$ for $10^{-6}$ in that case. For the Hessian-Free algorithm FoMAML, we fine-tuned $\lambda$ to meet the best performance. We set the learning rate of the outer and inner loops to $10^{-4}$ and 0.01, respectively, for 4-Conv. To maintain conciseness, we only report the results of SHOT$_r$ except for the comparison with MAML baseline. We conducted our experiments on a single A100 GPU. We implemented our method using torchmeta Deleu et al. [2019] library except for the implementation of LEO Rusu et al. [2018a], where we used the official implementation and embeddings.

## 5.1 Characteristics of SHOT

**SHOT Outperforms the Baselines on Benchmarks and Cross-Benchmark Settings** Our method, SHOT, outperforms the corresponding baseline on all benchmarks, as shown in Table 2. Both SHOT$_r$ and SHOT$_p$ outperform the baseline, regardless of whether the benchmark is general or fine-grained. Additionally, SHOT generalizes well to cross-benchmark settings, as demonstrated in Table 3, where it improves the performance of the baseline. Also, we can see that this result is independent of the backbone architecture, as shown in Table 4.

Table 4: Test accuracy % of ResNet-12. The values in parentheses indicate the number of shots. The better accuracy between the baseline and SHOT is bold-faced.

| meta-train | miniImageNet | | |
|---|---|---|---|
| meta-test | miniImageNet | tieredImageNet | Cars |
| MAML (1) | $49.47 \pm 0.43$ | $54.88 \pm 1.07$ | $\mathbf{32.74} \pm 0.48$ |
| MAML + SHOT$_r$ (1) | $\mathbf{51.11} \pm 0.32$ | $\mathbf{55.14} \pm 0.63$ | $32.66 \pm 0.27$ |
| MAML (5) | $69.12 \pm 0.08$ | $71.60 \pm 0.08$ | $51.99 \pm 0.25$ |
| MAML + SHOT$_r$ (5) | $\mathbf{69.74} \pm 0.16$ | $\mathbf{71.71} \pm 0.39$ | $\mathbf{52.16} \pm 0.28$ |

**SHOT works with Hessian-Free and One-Step algorithms** As many GBML algorithms are based on Hessian-Free algorithms which only exploit gradients and One-Step algorithms which only exploit one step in the inner loop Finn et al. [2017], Nichol et al. [2018], Oh et al. [2020], Raghu et al. [2019], it is important to show that our method also works with these algorithms. As shown in Table 5, our method improves the performance of the baseline on both Hessian-Free and One-Step algorithms. For One-Step algorithms, we set $\theta_t$ as the original model and $\theta_r$ as the 2-step optimized model. This is practically useful since many algorithms exploit only one step in the inner loop. Notably, our method is applied only in the training phase, so it has the same inference cost as the baseline. Also, we only used first-order optimization for the meta-optimizer to ensure a fair comparison when adding SHOT to FoMAML. For ANIL Raghu et al. [2019] and BOIL Oh et al. [2020], we conducted experiments with the same settings from each baseline paper, and we bring the results from Oh et al. [2020] for comparison.

**SHOT enables faster convergence**. Our method, SHOT, achieves faster convergence compared to the baseline, as demonstrated in Figure 3, where we plot the validation accuracy against the training

Table 5: Test accuracy (%) of 4-conv network on benchmark data sets. The values in parentheses indicate the number of shots. The better accuracy between the baseline and SHOT is bold-faced. Baselines in the table corresponds to either one-step algorithm (which uses one step optimization in the inner loop) or Hessian-Free algorithm (an algorithm which only uses gradients in the outer loop).

| Domain | General(Coarse-grained) | | Specific (Fine-grained) | |
|---|---|---|---|---|
| Dataset | miniImageNet | tieredImageNet | Cars | Cub |
| FoMAML (1) | $47.70 \pm 0.56$ | $47.72 \pm 0.76$ | $\mathbf{49.08} \pm 1.46$ | $60.18 \pm 0.45$ |
| FoMAML + SHOT$_r$ (1) | $\mathbf{48.49} \pm 0.29$ | $\mathbf{47.91} \pm 0.59$ | $48.59 \pm 0.72$ | $\mathbf{60.54} \pm 0.29$ |
| ANIL (1) | $47.82 \pm 0.20$ | $\mathbf{49.35} \pm 0.26$ | $46.81 \pm 0.24$ | $57.03 \pm 0.41$ |
| ANIL + SHOT$_r$ (1) | $\mathbf{48.02} \pm 0.38$ | $48.29 \pm 0.93$ | $\mathbf{47.82} \pm 1.48$ | $\mathbf{57.31} \pm 0.89$ |
| BOIL (1) | $49.61 \pm 0.16$ | $48.58 \pm 0.27$ | $\mathbf{56.82} \pm 0.21$ | $61.60 \pm 0.57$ |
| BOIL + SHOT$_r$ (1) | $\mathbf{50.36} \pm 0.42$ | $\mathbf{50.36} \pm 0.49$ | $56.22 \pm 1.96$ | $\mathbf{62.43} \pm 0.30$ |
| FoMAML (5) | $64.49 \pm 0.46$ | $65.25 \pm 0.64$ | $67.99 \pm 1.20$ | $73.18 \pm 0.18$ |
| FoMAML + SHOT$_r$ (5) | $\mathbf{64.58} \pm 0.06$ | $\mathbf{65.92} \pm 0.22$ | $\mathbf{68.02} \pm 1.00$ | $\mathbf{73.89} \pm 0.18$ |
| ANIL (5) | $63.04 \pm 0.42$ | $65.82 \pm 0.12$ | $61.95 \pm 0.38$ | $70.93 \pm 0.28$ |
| ANIL + SHOT$_r$ (5) | $\mathbf{64.40} \pm 0.75$ | $\mathbf{66.19} \pm 0.13$ | $\mathbf{61.00} \pm 0.90$ | $\mathbf{71.71} \pm 1.34$ |
| BOIL (5) | $\mathbf{66.45} \pm 0.37$ | $\mathbf{69.37} \pm 0.12$ | $75.18 \pm 0.21$ | $75.96 \pm 0.17$ |
| BOIL + SHOT$_r$ (5) | $66.34 \pm 0.21$ | $69.20 \pm 0.25$ | $\mathbf{75.25} \pm 0.14$ | $\mathbf{76.77} \pm 0.51$ |

epoch. To understand why this occurs, we plotted averaged cosine similarity between the overall parameter difference with each inner gradient step in the inner loop, as shown in Figure 3. We found that overall similarity increases in both cases of with and without SHOT, and the similarity is highly correlated with the validation accuracy. SHOT enforces high similarity from the beginning, resulting in faster convergence with better generalization ability.

**Applying other distance metrics**. We used KL divergence as a distance metric in SHOT, but it is possible to use other metrics as well. For example, we could use cross-entropy as a distance metric from a probability perspective, or we could use L2 distance between parameters as a distance metric. We conducted experiments with different distance metrics, and for L2 distance we divided the L2 distance by the square root of the number of parameters to account for the initialization scheme Kumar [2017]. For the cross-entropy loss, we used the same $\lambda$ (balancing term for Eq. 4) settings as for KL divergence. The results are shown in Table 6. We observed that SHOT outperformed the baseline for every distance metric. This supports our hypothesis that we can improve performance and dynamics by suppressing the effect of the Hessian along the optimization trajectory.

## 6 Conclusion

In this paper, we claim that GBML implicitly suppresses the Hessian along the optimization path. We also demonstrate that we can improve the performance of GBML by explicitly enforcing this implicit prior. To achieve this, we propose a novel method, SHOT (Suppressing the Hessian along Optimization Trajectory). Although it deals with curvature of the loss surface, it has the same computational order compared to other GBML methods. More specifically, it adds a negligible computational overhead to the baseline and costs the same at inference time. We show that SHOT outperforms the baseline and is algorithm- and architecture-independent. We believe that our method can be applied to other GBML methods in a plug-and-play manner.

Table 6: Test accuracy % of 4-conv using SHOT with various distance metrics. The values in parentheses indicate the number of shots. The better accuracy between the baseline and SHOT is bold-faced.

| meta-train | miniImageNet | | |
|---|---|---|---|
| meta-test | miniImageNet | tieredImageNet | Cars |
| Baseline (1) | $47.88 \pm 0.55$ | $51.84 \pm 0.24$ | $34.41 \pm 0.47$ |
| KL-Divergence (1) | $47.97 \pm 0.71$ | $51.68 \pm 0.68$ | $34.03 \pm 1.11$ |
| Cross-Entropy (1) | $49.07 \pm 0.14$ | $53.83 \pm 0.14$ | $34.54 \pm 0.31$ |
| L2 Distance (1) | $\mathbf{50.89} \pm 0.06$ | $\mathbf{54.63} \pm 0.25$ | $\mathbf{35.75} \pm 0.29$ |
| Baseline (5) | $64.81 \pm 1.63$ | $67.96 \pm 1.22$ | $46.57 \pm 0.53$ |
| KL-Divergence (5) | $\mathbf{66.86} \pm 0.58$ | $\mathbf{71.77} \pm 0.42$ | $48.42 \pm 0.72$ |
| Cross-Entropy (5) | $66.34 \pm 0.35$ | $69.47 \pm 0.35$ | $\mathbf{48.86} \pm 0.71$ |
| L2 Distance (5) | $66.85 \pm 0.44$ | $69.96 \pm 0.27$ | $48.40 \pm 0.12$ |

## 7 Acknowledgements

This work was supported by NRF grant (2021R1A2C3006659) and IITP grants (2021-0-01343, 2022-0-00953), all of which were funded by Korean Government (MSIT).

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

# A Proof of (3)

Due to fundamental theorem of calculus,

$$L(\theta^{k+1}) - L(\theta^k) = \int_C \nabla L(\theta) \cdot d\theta = \int_0^1 \nabla L(\theta(t)) \cdot v(t)dt, \tag{5}$$

where $C$ is a trajectory whose start and end points are $\theta^k$ and $\theta^{k+1}$. In GD setting, because $\theta^{k+1} = \theta^k - \alpha \nabla L(\theta^k)$ for some learning rate $\alpha > 0$, we can think of the straight line trajectory joining $\theta^k$ and $\theta^{k+1}$. In this case, the velocity vector becomes $v(t) = -\alpha \nabla L(\theta^k)$ and

$$L(\theta^{k+1}) - L(\theta^k) = -\alpha \int_0^1 \nabla L(\theta(t)) \cdot \nabla L(\theta^k)dt \tag{6}$$

where $\theta(0) = \theta^k$, $\theta(1) = \theta^{k+1}$ and $\theta(t) = (1-t)\theta(0) + t\theta(1)$.

By Taylor series expansion it becomes

$$\nabla L(\theta(t)) \approx \nabla L(\theta^k) + H(\theta^k)(\theta(t) - \theta^k) = (I - \alpha t H(\theta^k))\nabla L(\theta^k) \tag{7}$$

and combining Eq.(6) and Eq.(7) proves Eq.(3).

# B SHOT is robust to hyperparameter settings

Table 7: Test accuracy % of 4-conv using SHOT with different $\lambda$ values. The values in parentheses indicate the number of shots. The better accuracy between the baseline and SHOT is bold-faced.

| size of $\lambda$ | miniImageNet (5) |
|---|---|
| 0 (baseline, MAML) | $64.81 \pm 1.63$ |
| 0.1 (SHOT$_r$) | $\mathbf{66.86 \pm 0.58}$ |
| $10^{-2}$ | $65.97 \pm 0.93$ |
| $10^{-3}$ | $66.85 \pm 0.47$ |
| $10^{-4}$ | $66.08 \pm 0.36$ |
| $10^{-5}$ | $66.81 \pm 0.28$ |

Table 7 shows the performance of SHOT with various hyperparameter settings. We conducted experiments with different learning rates of $\lambda$, and in all cases, SHOT improved the performance of the baseline. This suggests that SHOT is robust to hyperparameter settings.

# C Another viewpoint of ANIL and BOIL

In this section, we reconcile the opposite opinions of feature reuse versus feature adaptation in gradient-based meta-learning (GBML) with our hypothesis that reducing the impact of the Hessian in the inner loop can improve performance.

ANIL, proposed in Raghu et al. [2019], argues that feature reuse is key in the inner loop, where the feature remains invariant while only the decision boundary is adapted. On the other hand, BOIL, proposed in Oh et al. [2020], argues that feature adaptation is key in the inner loop, where the feature is adapted while the decision boundary remains invariant.

To test their arguments, ANIL and BOIL proposed two algorithms. ANIL freezes the encoder and only updates the head in the inner loop, while BOIL freezes the head and only updates the encoder in the inner loop. The problem is that both algorithm shown good performance thereby both arguments look persuasive despite they argue exactly in the opposite ways.

Our hypothesis, which suggests that the outer loop implicitly suppresses the Hessian along the optimization trajectory, can reconcile the arguments of both ANIL and BOIL. This is because our hypothesis implies that the model acts linearly in the inner loop. ANIL and BOIL can be interpreted as algorithms that enforce linearity in the inner loop by restricting parameters and reducing the number of non-linear components between layers.

ANIL freezes the encoder and only updates the head in the inner loop, reducing the number of non-linear components in the inner loop. This enforces linearity in the inner loop, as the only non-linearity is the loss function. ANIL achieves better performance than MAML in 1-step optimization, as it is more powerful at 1-step optimization, which views the model as linear.

BOIL freezes the head and only updates the encoder in the inner loop, reducing the number of non-linear components in the inner loop. By applying BOIL, the gradient norm is predominant in the last layer of the encoder, making it a variant of ANIL that updates only the penultimate layer. This layer has much stronger performance, as it can change the feature while maintaining the linearity of the model. Table 14 of Oh et al. [2020] shows a boosted performance when all but the penultimate layer is not frozen. By explicitly enforcing linearity in the inner loop, BOIL achieves improved performance.

## D   GBML is a variant of Prototype Vector method

In this section, we provide a novel viewpoint of GBML, that GBML (Gradient-Based Meta Learning) is a varient of MBML (Metric-Based Meta Learning). This viewpoint relies on the **linearity assumption**. *i.e.*, the effect of the Hessian along the optimization trajectory is zero, thereby the model act as linear in the inner loop.

Suppose there exists a meta-learning model that satisfies the linearity assumption in the inner loop, then classifying a new classification task with a task-specific function $f(\cdot|\theta^\star)$ after an inner loop is equivalent to creating a prototype vector for each class on a specific feature map and classifying the input as the class of the most similar prototype vector.

The proof starts by defining the prototype vector at first.

**Prototype Vector** We define a prototype vector $V_c$ for class $c$ in an $N$-way $K$-shot classification task formally as

$$V_c = \sum_{i=1}^{N} \sum_{j=1}^{K} \beta_{ij} \varphi_c(X_{ij}), \quad c \in \{1, \cdots, N\}, \tag{8}$$

where $X_{ij}$ is the $j$-th input sample for the $i$-th class, $\varphi_c(\cdot) \in \mathcal{H}$ is a class-specific feature map and $\beta_{ij}$ indicates the importance of $X_{ij}$ for constituting the prototype vector $V_c$.

In other words, there exists a feature map $\varphi_c$ for each class $c$, and the support set is mapped to the corresponding feature map and then weighted-averaged to constitute the prototype vector of the corresponding class. At inference time, the classification of a given query $X$ is done by taking the class of the most similar prototype vector as follows:

$$\hat{c} = \arg \max_c \langle V_c, \varphi(X) \rangle. \tag{9}$$

Here, $\varphi : \mathcal{X} \to H$ is a non-class-specific mapping. We can also rewrite the prototype vector using $\varphi$ and by defining a projection $P_c : \mathcal{X} \to \mathcal{X}$ as

$$P_c(X) = \begin{cases} X & \text{if } y(X) = c, \\ \nu \in \mathcal{N}(\varphi), & \text{if } y(X) \neq c \end{cases} \tag{10}$$

where $y(X)$ is the ground truth class of $X$ and $\mathcal{N}$ is the null space of $\varphi$ *i.e.*, $\varphi(\nu) = 0$.

Then by defining $\varphi_c \triangleq \varphi \circ P_c$ and $\beta_{ij} \triangleq \frac{1}{K}$, it becomes

$$V_c = \frac{1}{K} \sum_{j=1}^{K} \varphi(X_{cj}). \tag{11}$$

**SGD in the inner loop** If GBML satisfies the hypothesis of linearity in the inner loop, $f$ is locally linear in $\theta$ in an inner loop. More specifically, there exists an equivalent feature map $\varphi_c : \mathcal{X} \to \mathcal{H}$

which satisfies $f_c(\cdot|\theta_c) = \langle\theta_c, \varphi_c(\cdot)\rangle$ for every $x \in \mathcal{X}$ where $f(\cdot|\theta) = [f_1(\cdot|\theta_1), \cdots, f_N(\cdot|\theta_N)]^T$. With the loss function $L(x, y|\theta) = D(s(f(x|\theta)), y)$ for some distance measure $D$ such as cross entropy, we can formulate the inner loop of $N$-way $K$-shot meta learning by SGD as

$$\theta_c^{k+1} = \theta_c^k - \alpha \sum_{i=1}^{N} \sum_{j=1}^{K} \frac{\partial L(X_{ij}, y(X_{ij})|\theta)}{\partial \theta_c} = \theta_c^k - \alpha \sum_{i=1}^{N} \sum_{j=1}^{K} \frac{\partial D}{\partial f_c} \varphi_c(X_{ij}), \quad (12)$$

since all samples in the support set are inputted in a batch of an inner loop.

Because the model is linear in the inner loop, the batch gradient does not change. Let $\beta_{ij} = -\frac{\partial D}{\partial f_c}|_{\theta_c^0, X_{ij}}$. Then after $t$ steps, by (8), the model becomes

$$\theta_c^t = \theta_c^0 + \alpha t \sum_{i=1}^{N} \sum_{j=1}^{K} \beta_{ij} \varphi_c(X_{ij}) = \theta_c^0 + \alpha t V_c. \quad (13)$$

At the initialization step of an inner loop, there is no information about the class, even the configuration order of the class, because the task is randomly sampled. If so, the problem is solved in the inner loop. For example, if a class *Dog* is allocated to a specific index such as *Class 3*. There is no guarantee that it will have the identical index the next time the class *Dog* comes in. Thus, at a meta-initialization point $\theta^0$, the scores for different classes would not be much different, *i.e.*, $f_i(x|\theta_i^0) \simeq f_j(x|\theta_j^0)$ for $i, j \in [1, \cdots N]$.

Considering the goal of classification is achieved through relative values between $f_i(X)$'s, the value at the initialization point does not need to be considered significantly. Therefore

$$\arg\max_c f_c(X) = \arg\max_c \langle\theta_c^t, \varphi(X)\rangle = \arg\max_c \langle\theta_c^0 + \alpha t V_c, \varphi(X)\rangle \sim \arg\max_c \langle V_c, \varphi(X)\rangle \quad (14)$$

So inner loop in GBML can be interpreted as making proptotype vector with given support set. $\square$

Table 8: Test accuracy % of 4-conv network on benchmark data sets. The values in parentheses indicate the number of shots. The best accuracy among different methods is bold-faced. To differentiate the notation, we have denoted $SHOT_3$ as a model that uses 3 optimization steps in $f_r$ and 1 optimization step in $f_t$, and $SHOT_6$ as a model that uses 6 optimization steps in $f_r$ and 3 optimization steps in $f_t$.

| meta-train | miniImageNet | | | Cars | | |
|---|---|---|---|---|---|---|
| meta-test | miniImageNet | tieredImageNet | Cars | Cars | miniImageNet | CUB |
| MAML (1) | $47.88 \pm 0.55$ | $\mathbf{51.84} \pm 0.24$ | $34.41 \pm 0.47$ | $47.78 \pm 0.99$ | $28.67 \pm 1.17$ | $30.95 \pm 1.41$ |
| MAML + $SHOT_3$ (1) | $47.97 \pm 0.71$ | $51.68 \pm 0.68$ | $34.03 \pm 1.11$ | $\mathbf{50.44} \pm 0.62$ | $\mathbf{30.19} \pm 0.50$ | $\mathbf{31.21} \pm 0.64$ |
| MAML + $SHOT_6$ (1) | $\mathbf{48.15} \pm 0.31$ | $51.67 \pm 0.73$ | $\mathbf{34.79} \pm 0.70$ | $49.89 \pm 0.17$ | $28.39 \pm 0.37$ | $30.83 \pm 0.26$ |
| MAML (5) | $64.81 \pm 1.63$ | $67.96 \pm 1.22$ | $46.57 \pm 0.53$ | $62.24 \pm 2.01$ | $37.23 \pm 1.95$ | $41.84 \pm 1.25$ |
| MAML + $SHOT_3$ (5) | $66.86 \pm 0.58$ | $69.48 \pm 0.17$ | $\mathbf{48.42} \pm 0.72$ | $\mathbf{69.08} \pm 0.31$ | $\mathbf{40.79} \pm 0.93$ | $\mathbf{43.46} \pm 0.89$ |
| MAML + $SHOT_6$ (5) | $66.27 \pm 0.23$ | $\mathbf{69.60} \pm 0.31$ | $45.83 \pm 1.82$ | $66.37 \pm 1.93$ | $39.35 \pm 0.74$ | $42.63 \pm 0.24$ |

# E   SHOT with more optimization step in the inner loop

In the main paper, we used only one step for the target model to improve computation efficiency. However, it's also important to test if SHOT works with more optimization steps in the inner loop. As a reference model, we set the number of optimization steps to 6, which is different from the main paper where we only used 3 steps (same as the baseline). As shown in Table 9, SHOT still performs better than the baseline even with more optimization steps in the inner loop.

# F   SHOT can act as a regularlizer

Regularlization techniques are widely used in many GBML algorithms Rajeswaran et al. [2019], Rusu et al. [2018a] and SHOT can also act as a regularizer. We replaced the regularization term of LEO Rusu et al. [2018a], a popular GBML algorithm that uses regularization in the inner loop, with

Table 9: Test accuracy % of LEO [Rusu et al., 2018a]. The values in parentheses are the number of shots. The better accuracy between the baseline and SHOT is bold-faced.

| Dataset | miniImageNet | tieredImageNet |
|---|---|---|
| LEO (1) | $60.24 \pm 0.02$ | $65.07 \pm 0.14$ |
| LEO + SHOT$_r$ (1) | $\mathbf{60.39} \pm 0.07$ | $\mathbf{65.26} \pm 0.10$ |
| LEO (5) | $75.27 \pm 0.13$ | $\mathbf{79.87} \pm 0.06$ |
| LEO + SHOT$_r$ (5) | $\mathbf{75.32} \pm 0.17$ | $79.84 \pm 0.14$ |

SHOT. The results, shown in Table 9, demonstrate that SHOT slightly enhances LEO's performance. Although the results are marginal, there is still room for improvement considering LEO works with highly-fine-tuned hyperparameters. This suggests the potential of SHOT as a plug-and-play regularization technique for GBML algorithms, whether it is highly-fine-tuned or not. Also, the role of SHOT as a regularizer explains why regularization techniques are effective in GBML. If we minimize the total transport distance in the inner loop, parameters do not move much even with a large learning rate, thereby the effect of the Hessian reduces.

## G   Algorithm

---

**Algorithm 1** Training Algorithm for GBML Model

---

```
Initialize parameter theta (the model)

while not reach max number of epochs:
    # Sample query set Q, support set S
    # Start of Inner loop
    Initialize reference model and target model with theta
    theta_r = theta
    theta_t = theta

    # Optimize theta_r with more (e.g. 3) inner loop steps step_r
    # and a smaller learning rate alpha
    for i in range(step_r):
        theta_r = theta_r - alpha * gradient(loss(theta_r, S), theta_r)

    # Optimize theta_t with less (e.g. 1) inner loop steps step_t
    # and a larger learning rate step_r / step_t * alpha
    theta_t = theta_t - step_r / step_t * alpha *
        gradient(loss(theta_t, S), theta_t)
    # End of Inner loop

    # Start of Outer loop
    # Calculate outer loss
    outer_loss = loss(theta_r, Q)
    # Reduce the distance between theta_r and theta_t using KL-divergence
    kl_loss = KL_divergence(target_distribution(theta_t, Q)
        , reference_distribution(theta_r, Q).detach())
    outer_loss += lambda * kl_loss
    # Use meta-optimizer to optimize parameter theta
    theta = theta - beta * gradient(outer_loss, theta)
    # End of Outer loop
# End of training loop
```

---

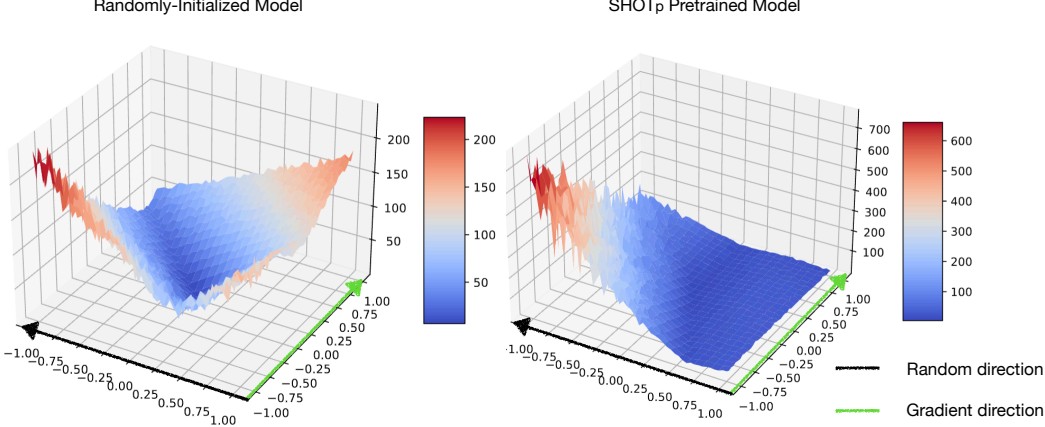

Figure 4: Comparison of loss surface between randomly initialized model and SHOT$_p$ initialized model. Showing that effect of the Hessian is suppressed is along the optimization trajectory.

# H  Loss surface actually gets linear along the optimization trajectory

Fig. 4 shows that our algorithm actually do what we expected. We compared between randomly-initialized model and SHOT$_p$ initialized model. Randomly-initialized model shows rough loss surface. The characteristic doesn't differ whether it is gradient's direction or not. However, SHOT$_p$ pretrained models shows smooth surface along the gradient direction, while other direction remains rough.

