# OpenReview forum: "SHOT: Suppressing the Hessian along the Optimization Trajectory for Gradient-Based Meta-Learning"
_NeurIPS.cc/2023/Conference — NeurIPS 2023 poster_

### Official Review · Reviewer_eLqV · 2023-07-06

**Soundness:** 2 fair
**Presentation:** 2 fair
**Contribution:** 3 good
**Rating:** 6
**Confidence:** 4

**Summary:**

The authors raise the hypothesis that gradient-based meta-learning implicitly suppresses the Hessian along the inner-loop’s optimization trajectory, and introduce a method (‘SHOT’) to leverage the trajectory of an already trained reference model to guide their target model’s optimization path based on this hypothesis.

**Strengths:**

* Interesting avenue of research and investigation
* Good motivation of the underlying mechanics and basis of the posed hypothesis, especially the importance of the Hessian in FSL settings in contrast to conventional many-shot deep learning
* Demonstrated versatility of proposed method by incorporating it into a range of different FSL-approaches, and tested across a range of few-shot setting and datasets


**Weaknesses:**

**TLDR**: While I do think the idea presented here is very interesting, the current form of this manuscript has severe weaknesses that should (and need to) be improved by the authors!

Major weaknesses:

**W1: Validity of some of the comparisons (to baselines)**
 - The main message of Table 1: What is the underlying reference model that the loss is applied to? (And is it applied to its initialization set of parameters?)
Why is the comparison done to a ‘randomly initialized’ model? I feel that adding a comparison to a meta-learnt baseline as well as the reference model would be more appropriate and insightful to see what ‘SHOT’ actually causes, and what is achieved already by any meta-trained (reference) model.
 - Evaluation for e.g. the one-step method is done with “baselines using 1 training step”, while SHOT uses a 2-step reference model and distills this (potentially much more powerful) information into a 1-step method. While I am not questioning the contribution of the authors to use such distillation, it would be important to state:
   - How well the underlying 2-step method actually performs (i.e. baseline with 2-opt steps);
   - How well does training the baseline with 2-steps of learning rate alpha, and evaluating it on 1-step with 2*alpha (no SHOT) perform?
- L.338: ‘faster convergence’ of the training procedure -> Isn’t the ‘basis of comparison’ misleading, since your model is distilling information from an already converged model for its optimization path and initialization point,  and thus has a much stronger training signal than conventional meta-training? (so the overall convergence to enable SHOT is actually the ‘original’ model + the SHOT-constraint model, isn’t it?)

**W2: Insufficient discussion of approach w.r.t to other works**
-  Relationship to distillation methods must be (briefly) discussed, since method very alike in terms of how objective is posed
-  Preconditioning methods in FSL/Meta-Learning must be discussed in more detail:
    - Lines108ff: “ *SHOT can be viewed as a variant of preconditioning […]* ” -> Yet, many recent methods are neither mentioned nor contrasted to/discussed -> See [e.g. 1,2,3]
    - Lines 109/110 state: “*unlike previous work, we define the condition of the meta-parameter based on the curvature of the loss surface*”; Note that the authors of [1] also use the curvature of the loss surface via an approximation of the Hessian for their preconditioning method
-> I’d like to see a discussion of:
What is the relation of the findings of this work to the findings in [1] (both are different in their perspective, [1] encouraging low condition number vs. your work encouraging low magnitude like SAM) – you nicely discuss the difference to SAM, so I feel this additional discussion would be a valuable and necessary addition to this paper

[1] Hiller et al., 'On Enforcing Better Conditioned Meta-Learning for Rapid Few-Shot Adaptation', NeurIPS 2022
[2] Simon et al., 'On modulating the gradient for meta-learning', ECCV 2020
[3] Park and Oliva, 'Meta-curvature', NeurIPS2019

**Further claims/points that need clarification**:
-> Please see the "Questions" section.


**Questions:**

In addition to the two main points raised in the weaknesses section, I'd like further clarification on the following points:

- Main method: Equation (4) to me essentially is a distillation loss, distilling a set of parameters for the student (‘target’) from a teacher (‘reference’) model;
   - How does the motivation of suppressing the Hessian here really apply? The loss is enforcing similarity between the parameters, how is it applied in practice? If the target model takes 3 steps and the reference model 6, is the set of target-parameters at step 1 compared to reference at step 2? Or how is this performed? (the stated lr (alpha) in line 226 somewhat indicates this via the fractionally-adjusted alpha, but it would be advantageous for the reader to point this out explicitly)
    - Also, does the Hessian really play a major role in this? If you are providing ‘optimal endpoints’ or at least ‘optimal directions’ for a large step-size via the other model, only gradient-direction (highly local) really matters, doesn’t it? Is the Hessian then really suppressed, or rather ignored?

- "Suppressing the Hessian" (name of algorithm) -> Is it possibility to actually compute the Hessian (potentially approximated version) for e.g. the small Conv4 network? It would be interesting to see whether it’s actually the case that the magnitude is small!  (It’s one of the main claims & motivations of the approach after all)


Some additional notes/hints:
 - Sec 5.1: “SHOT outperforms on all benchmarks” – might be wise to tone this down a bit, since the confidence intervals of almost all results are highly overlapping! (e.g. something like “performs better or on-par” might be more suited)
 - Heading 4.4 (and line 267) is an overstatement: Since shot is inflicting additional parameters and forward steps, stating that it “does not require ANY computational burden” is incorrect and should be modified appropriately (maybe akin to the wording in the introduction); It might be helpful to put further emphasis the difference between training overhead and inference time overhead (or potential saving) as done towards the end of this paragraph
 - Line 290: Lambda has not been introduced previously (to the best of my knowledge)
 - Formulation l.40 – 42: “instability caused by Hessian” -> I feel this is misleading and think the authors should clarify this -> The Hessian itself is simply a representation/measure of the curvature of the loss surface, but doesn’t itself cause anything. Instability will be due to highly non-smooth or curved loss-surface of the initial parameter space (which can be quantified via the Hessian)
 - I understand that this is purely for motivation, but the statement of ViT using 1e-6  is incorrect -> If you check the ViT paper, they state lr in the range of 1e-4 – 1e-3, and note that this is for ImageNet where significantly more parameter updates are performed due to dataset size
 - Conclusion section: I suggest reformulating the first sentence and removing the “we claim”, since at this point of the paper, you (want to) have proven/substantiated your claim.

> **UPDATE**: Updated rating to weak accept after clarification regarding the use of the term 'reference model' during rebuttal phase.


**Limitations:**

Limitations are not clearly stated, and the authors should improve upon this (include potential limitations that the authors are aware of).

---

> ### Author Rebuttal · Authors · 2023-08-08
>
> The main purpose of Table 1 is to highlight that 'Hessian-independence in the inner loop' is a crucial condition for GBML. We compared the performances of parameters that are randomly initialized and those pre-trained with SHOT (parameters trained 'ONLY' with the SHOT loss, which reinforces Hessian-independence in a self-supervised manner without explicit supervisory signals). This comparison aims to demonstrate that merely meeting the necessary condition significantly enhances the model's meta-learning ability.
>
> Concerning the evaluation of the one-step method, we consent on your opinion and experience on the baseline. The miniimagenet accuracies of SHOT are  62.65% for 5-shot and 47.9% for 1 shot in 4-conv setting(Without SHOT, it recorded 61.75% for 5 shot and 47.44% for 1 shot [1] (BOIL))
>
> We chose one-step and three-step optimization step for inference. This is because the original paper [2] (MAML) claimed that three optimization step is enough, and one-step methods are widely researched practically. For two-step, we conducted experiments on miniimagenet, and the setting is the same for other optimization.  In this case, both 5 shot and 1 shot showed betterr performance than the baselines. For 1 shot, SHOT performed 49.01% whereas baseline showed 47.13%, also, for 5 shot, SHOT obtained 65.56% which is slightly better than 65.26% of the baseline.
>
> We apologize for the confusion regarding the term 'reference' in L.338.) We meant it to refer to a model that is less influenced to the Hessian compared to the target model. Both the target model and the reference model share the same meta-parameter within the same task. The only difference lies in the number of optimization steps and learning rate in the inner loop. Which means, we do not use any pretrained reference model during training. We will provide a pseudo code to clarify this distinction.
>
> We appreciate your mention of recent advancements in preconditioning methods in meta-learning and connection to distillation. We noticed that ‘reference model’ can be view as a teacher in broad range. We will include brief explanation on this. Also, our approach, SHOT, does employ a form of preconditioning, but it's fundamentally different from the episode-dependent methods discussed in [3][4].
> Unlike these methods, SHOT seeks a 'global' initialization point 'along the inner loop' where the optimization trajectory is independent of the Hessian. This contrasts with other methods that attempt to satisfy a specific condition in the entire Hessian, which involves wide exploration on the local surface.
>
> This distinction allows SHOT to operate under conditions relaxed in dimensionality - we consider dimension of $p$ (the number of parameters), rather than a $p \times p$ matrix, reducing the need for extensive computational resources [3][5] or specific architectural adaptations [4]. This highlights the novelty of our work.
>
> Lastly, in relation to the suggested link between low condition numbers and our work, we acknowledge that while those methods deal with the entire Hessian, SHOT specifically targets the effectiveness of the Hessian along the optimization trajectory.
>
> [1] Oh, Jaehoon, et al. "Boil: Towards representation change for few-shot learning." arXiv preprint arXiv:2008.08882 (2020).
>
> [2] Finn, Chelsea, Pieter Abbeel, and Sergey Levine. "Model-agnostic meta-learning for fast adaptation of deep networks." International conference on machine learning. PMLR, 2017.
>
> [3] Park and Oliva, 'Meta-curvature', NeurIPS 2019
>
> [4] Simon et al., 'On modulating the gradient for meta-learning', ECCV 2020
>
> [5] Hiller et al., 'On Enforcing Better Conditioned Meta-Learning for Rapid Few-Shot Adaptation', NeurIPS 2022
>
>
> Regarding the questions,
>
> 1. As mentioned earlier, we do not require a 'pre-trained' reference model. Nonetheless, we agree that the structure of the loss itself mirrors that of distillation loss. Therefore, we would like to reference previous distillation works in the related works section of our paper.
> Response to your question about the reference model taking 6 steps when the target model takes 3 steps as discussed in Sec E, the learning rate of the reference model becomes half that of the target model. Additionally, we detach the reference model’s logit in $L_{SHOT}$.
> 2. We want to clarify that we do not advocate for 'Suppressing the Whole Hessian.' As described in Eq.3, our primary concern is the Hessian ALONG the inner loop, which presents a more relaxed condition.
> 3. We agree with your observation that many values lie between the standard deviations. We consent that there is a need to tone down as you suggested. However, it is essential to note that we reported the sample standard deviation, not the mean standard deviation. Applying the sample std, most of the values exceed the variance margin.
> 4. As previously stated, we do not require a pre-trained model. Therefore, in SHOT$_r$, no additional backpropagation is needed.
> 5. We appreciate you pointing out the need for clarity regarding $\lambda$. $\lambda$ indeed serves as the balancing term for Eq.4, and the outer loop loss becomes Eq.2 + $\lambda$ Eq.4. We will include this information in the paper.
> 6. Thank you for pointing out the need to change the learning rate term in regard to ViT. We will make this adjustment.
>
> Regarding your final point, because we did not include a rigorous proof of our claim, we  keep the original sentence.
>
> We hope these responses and the pseudo code below provide the clarification you were seeking and appreciate your insightful questions.

---

> > ### Comment · Reviewer_eLqV · 2023-08-13
> >
> > I'd like to thank the authors for their detailed answers and clarifications.
> >
> > I did indeed misinterpret the use of the term 'reference model' as outlined above, and thank the authors for clarifying this point and providing the pseudo-algorithm in the global rebuttal.
> >
> > The provided visualization is equally very insightful, and I recommend to include it into the final version of the paper (at least supplementary if no space).
> >
> > I encourage the authors to improve their manuscript regarding clarity to avoid potential confusion (which has also been commented on by other reviewers) and to include a slightly broader discussion of related work (as they have done in this rebuttal), since I think it will further elevate their specific contribution and place it well within other works considering the Hessian in this context.
> >
> > That said, my major concerns have been addressed and I change my rating to recommend a weak accept.

---

> > > ### Author Response · Authors · 2023-08-16
> > >
> > > Thank you for your insightful feedback.
> > >
> > > In response to your suggestions, we will enhance the paper by adding visualizations to clearly depict our intentions with the Hessian and to demonstrate the computational benefits of our algorithm. We will also include the pseudo-code to further elucidate our approach. Additionally, we recognize the gaps in our related work section and will incorporate the recent papers relevant to our study post-rebuttal.
> > >
> > > Again, Your feedback is invaluable, and we deeply appreciate it.

---

### Official Review · Reviewer_GyHa · 2023-07-07

**Soundness:** 3 good
**Presentation:** 2 fair
**Contribution:** 3 good
**Rating:** 6
**Confidence:** 4

**Summary:**

This paper proposes a hypothesis: _gradient-based meta-learning (GBML) implicitly suppresses the Hessian along the optimization trajectory in the inner loop_. Based on the hypothesis, proposes an algorithm that **implicitly suppress the hessian** by minimizing the distance between the target (trained with fewer steps within inner-loop) and reference model (trained with more steps within inner-loop). The idea is that **a model trained with smaller learning rate will have less distortion due to the hessian**. The proposed algorithm is algorithm- and model- agnostic and improves performance.

**Strengths:**

- Interesting idea that suppressing hessian.
- Impressive performance improvements.

**Weaknesses:**

- In Sec 4. some explanation is missing or confusing. Text can be more clearly written.
    - In my understanding, reference model is "less" (stated as "not" in L208) influenced by the Hessian, because of smaller learning rate, not increased the number of optimization steps. L97-99 states smaller learning rate makes hessian less affect the convergence.
    - $D(\theta_t, \theta_r)$ is a distance metric, but using KL divergence as a distance metric is confusing. I guess $D$ is not necessarily a distance metric.
    - How gradient from $L_{SHOT}$ is defined? Is $\theta_r^R$ detached or does it participate in gradient computation?
    - In table 1. how SHOT-init (or lower-distortion-initialization) is obtained? Are Random/SHOT-init meaning that they are trained w/o or w/ SHOT loss? Or is this relevant to $SHOT_p$?
    - What's the motivation of using model trained only with SHOT loss as meta-meta-initializer? If it starts with the initialization, then suppressed hessian property is maintained without SHOT loss?
    - Which parameter, $\theta_r$ or $\theta_t$, is used for MAML loss?
    - "$\theta_t$ does not require additional backprop." means that it reuses the gradient at the 1st step of $\theta_r$ but using different lr?

- $\lambda$ appears in Sec. 5, but not defined. (Maybe weights of SHOT loss?)


**Questions:**

 - How gradient from $L_{SHOT}$ is defined? Is $\theta_r^R$ detached or does it participate in gradient computation?
 - In table 1. how SHOT-init (or lower-distortion-initialization) is obtained? Are Random/SHOT-init meaning that they are trained w/o or w/ SHOT loss? Or is this relevant to $SHOT_p$?
- What's the motivation of using model trained only with SHOT loss as meta-meta-initializer? If it starts with the initialization, then suppressed hessian property is maintained without SHOT loss?
- Which parameter, $\theta_r$ or $\theta_t$, is used for MAML loss?

---

> ### Author Rebuttal · Authors · 2023-08-08
>
> Thank you for pointing out areas that need improvement. I'll provide the following responses to your points:
> Regarding the motivation drawn from Gradient-Flow, we acknowledge that our motivation could be clearer. The motivation behind using fewer epochs to reduce the loss $L$ and increasing the learning rate is essentially the same, as we explain through gradient flow. To avoid confusion, we plan to modify the wording to terms like 'fast-adaptation' and 'slow-adaption', emphasizing that a slower model evolution implies less influence from the Hessian.
>
> Also, you mentioned that D is not a necessary a distance metric. You are correct. KL divergence is not a distance metric in a strict sense since it's not symmetric. It can, however, serve as a pseudo-distance between models. We will replace the term `metric’ with a more appropriate term in the revised manuscript.
>
> We appreciate your observation about $\lambda$ in Sec. 5. $\lambda$ indeed serves as the balancing term for Eq.4, resulting in the outer loop loss of Eq.2 + $\lambda$ Eq.4. We will include this clarification in the revised manuscript.
>
> Regarding questions:
> 1. In L(SHOT), $\theta_r$ is indeed detached from gradient computation.
> 2. In Table 1, we trained SHOT$_r$ without the outer loss that forces the model to achieve high performance on the query set. As stated in L247, this approach is vulnerable to homogeneous corruption, so we used early stopping with 3 epochs. However, it may achieve higher performance if trained with SHOT$_p$. More details are discussed in the common response.
> 3. The motivation for using a model trained only with SHOT loss as the meta-meta-initializer is that around the initial point $\theta_0$ obtained by SHOT$_p$, the loss surface tends to be more linear than the region near a randomly initialized point. This makes meta-learning easier with the learned meta-initialization point.
> 4. For the MAML loss, we use $\theta_r$. (which means we use 3 steps in the inner loop, This is consistent across all methods in 4conv for inference. i.e., Same architecture, same inner loop setting except one-step optimization scheme.)
> 5. Yes, "$\theta_t$ does not require additional backprop" means that it reuses the gradient at the first step of $\theta_r$, but uses a different learning rate. Although it is not explicitly written in the code, functionally, it's equivalent.
>
> Besides, I also want to point out that the term ‘reference’ is used to indicate that it is less influenced by the Hessian ‘compared to’ the Target model. Both models share the same meta-parameter (starts from the same initialization $\theta_0$).

---

> > ### Comment · Reviewer_GyHa · 2023-08-20
> >
> > Thanks to the authors for answering my questions. Most of my questions are well-addressed.

---

### Official Review · Reviewer_Tzy2 · 2023-07-13

**Soundness:** 3 good
**Presentation:** 3 good
**Contribution:** 3 good
**Rating:** 6
**Confidence:** 3

**Summary:**

Authors consider meta learning task and formulate the hypothesis regarding property, common for gradient based approaches: they suppress Hessian implicitly due to the big step sizes in inner loop which works for functions, which are linear on a path to task-specific optimum. Proposed method does this suppression explicitly by changing loss function or adding regulariser to it, which computation is cheap and which can be incorporated in other GBML methods

**Strengths:**

The hypothesis formulated is natural, proposed approach is easy to combine with other approaches, empirical study' methodology is proper, experiments on several practically important tasks are presented.

**Weaknesses:**

In fact, I do not see significant advantage in using the proposed approach, because most of scores in tables, which should indicate the advantage of the approach, lie within ± std interval of other scores, which makes advantage statistically insignificant. If this impression doesn't correspond to authors' observations, they probably should present them in more friendly way, emphasizing the cases where advantage is statistically significant together with percentage of this advantage.

**Questions:**

No questions in addition to the recommendation given above.

**Limitations:**

Everything is okay.

---

> ### Author Rebuttal · Authors · 2023-08-08
>
> Thank you for your valuable comment. Your point about the statistical significance of our results is well taken. We want to clarify that what we reported in our tables after $\pm$ is the standard deviation among three differently seeded experiments. You seem to be saying that the standard deviation margins overlap, but what we reported is a sample standard deviation. Therefore, if we are to perform a t-test, it should actually be divided by the square root of the number of seeds.
>
> For instance, if we look at Table 2's mini-imagenet (5), when we divide the STD by the square root of 3, we observe a statistically significant performance improvement which does not overlap within the significance margin. We will clarify these details in our revised manuscript. Thank you again for your constructive feedback.

---

> > ### Comment · Reviewer_Tzy2 · 2023-08-18
> >
> > Dear authors, thank you for this clarification, I'm sure that it will make your result more clear for readers! I consider my overall score correct anyway, because I cannot see that practical significance of this novel approach is really great, it is only comparable and competitive to other approaches. However, I appreciate the direction of authors research itself and recommend to continue this work by enriching it theoretically and practically. This approach is perspective, but at the moment it requires more work on it.

---

### Official Review · Reviewer_d83w · 2023-07-28

**Soundness:** 3 good
**Presentation:** 3 good
**Contribution:** 3 good
**Rating:** 5
**Confidence:** 3

**Summary:**

This paper introduces an algorithm called SHOT (Suppressing the Hessian along the Optimization Trajectory) for gradient-based meta-learning (GBML). The authors hypothesize that GBML implicitly suppresses the Hessian along the optimization trajectory in the inner loop. Based on this hypothesis, they propose the SHOT that minimizes the distance between the parameters of the target and reference models to suppress the Hessian in the inner loop. This versatile approach, while without additional computation cost, proves to be effective in few-shot classification tasks.


**Strengths:**

1. The paper introduces a novel perspective on GBML by highlighting the potential benefits of suppressing the Hessian along the optimization trajectory, which can lead to performance improvements in meta-learning models.

2. The proposed SHOT algorithm is simple, effective, and versatile, with the potential to be applied to any GBML baseline, irrespective of the specific algorithm and architecture in use. This broad applicability increases its potential usefulness in the field of meta-learning.

**Weaknesses:**

1. While the authors provide empirical evidence to support their claims, it would be beneficial to supplement this with theoretical guarantees or additional concrete evidence to lend further weight to their arguments.

2. The paper would benefit from a wider range of experimental results, specifically those pertaining to few-shot regression tasks or reinforcement learning, in order to validate the effectiveness of SHOT across various tasks.



**Questions:**

Can the authors provide visualizations of the landscape to support their claims about the suppression of the Hessian along the optimization trajectory?

**Limitations:**

The paper lacks a discussion regarding the limitations and potential adverse societal impacts of the proposed model in the main paper. I strongly suggest adhering to the conference rules by addressing this crucial aspect.

---

> ### Author Rebuttal · Authors · 2023-08-08
>
> Thank you for your detailed feedback and thoughtful questions regarding our paper.
>
> In response to your request for additional theoretical backing, I think appendix C and D would give some theoretical background on your question. We have stated the connection between GBML and MBML (metric-based meta-learning) based on the linearity assumption of the loss surface in Appendix D. We have also provided another perspective of ANIL [1] and BOIL [2] in Appendix C. As explicitly stated in lines 511-514 of Appendix, a meta-learning model that satisfies the linearity assumption in the inner loop is able to classify a new classification task using a task-specific function $f(·|\theta^*)$ after the inner loop. This process equates to creating a prototype vector for each class on a specific feature map and then classifying the input as the class of the most similar prototype vector.
>
> Regarding your suggestion to include a broader range of experimental results, we acknowledge that incorporating tasks such as few-shot regression and reinforcement learning would enhance the paper. While our current references primarily focus on few-shot classification, we also agree that our architecture-independent algorithm can effectively extend its applicability to broader tasks including the ones you suggested. However, due to limited rebuttal period, we are not able to provide additional experimental results here and plan to supplement our material to cover these additional areas after the rebuttal period.
>
> Your suggestions about providing visualizations to support our assertions about the suppression of the Hessian along the optimization trajectory is very pertinent. We have indeed performed such visualizations, which resulted in proving our claim. It only suppressed the Hessian along the inner loop. The figure can be found in a separate one-page PDF file attached during the rebuttal which will be included in the supplementary material.  Again, your feedback has been crucial in guiding the enhancement of our work, and we appreciate your constructive critique.
>
> [1] Raghu, Aniruddh, et al. "Rapid learning or feature reuse? towards understanding the effectiveness of maml." arXiv preprint arXiv:1909.09157 (2019).
>
> [2] Oh, Jaehoon, et al. "Boil: Towards representation change for few-shot learning." arXiv preprint arXiv:2008.08882 (2020).

---

> > ### Comment · Reviewer_d83w · 2023-08-14
> > **Response to rebuttal**
> >
> > Thanks to the authors for providing explanations to address my concerns. However, I'm still curious whether the proposed algorithm might work well in other tasks, such as few-shot regression and reinforcement learning. While the authors mentioned a limited rebuttal period, I believe these experiments do not require significant computation, so could have been conducted within the period. Thus, I want to keep my initial score for this paper.

---

> > > ### Author Response · Authors · 2023-08-17
> > > **Regarding performance on RL and regression**
> > >
> > > Sorry for late response, it took some time as our code is based on torchmeta [1], which is only for few-shot classification. We implemented few-shot regression [2] and rl [3] using different codes as bases.
> > >
> > > For few-shot regression, we remained the settings of the original code (five-step), and used a target model as 2-step optimized model. In this case, MSE is **2.70** $\pm$ **0.10** for the original code, and MSE = **2.20** $\pm$ **0.08** for ours. which is a notable increase of performance.
> > >
> > > For reinforcement learning, we also used the default settings of the original code in [3] and just ran it without any modification of parameters. As the original code's inner optimization number is 1, we used reference's optimization number as 2, which is the same setting as Table.5 in our paper. We reduced KL divergence which is generated in validation episodes. The environment is halfcheetah, maml-trpo. In this case, we have seen a significant increase of maximum reward on episodes during training, 11 to 62. However, it increased instability. We conjecture that this is because the original paper [4] have adopted some additional techniques to deal with this instability, which is out of the scope of MAML. i.e., adopted some details to make MAML work, which we did not in this experiment. We plan to provide the corresponding code and settings.
> > >
> > >
> > > [1] Deleu, T., Würfl, T., Samiei, M., Cohen, J. P., & Bengio, Y. (2019). Torchmeta: A meta-learning library for pytorch. arXiv preprint arXiv:1909.06576.
> > >
> > > [2] Nguyen, C. (2022). Few_shot_meta_learning [GitHub repository]. GitHub. https://github.com/cnguyen10/few_shot_meta_learning
> > >
> > > [3] Arnold, S. M., Mahajan, P., Datta, D., Bunner, I., & Zarkias, K. S. (2020). learn2learn: A library for Meta-Learning research. arXiv preprint arXiv:2008.12284.
> > >
> > > [4] Finn, Chelsea, Pieter Abbeel, and Sergey Levine. "Model-agnostic meta-learning for fast adaptation of deep networks." International conference on machine learning. PMLR, 2017.

---

> > > > ### Comment · Reviewer_d83w · 2023-08-17
> > > > **Response to rebuttal**
> > > >
> > > > I appreciate the additional experimental results provided by the authors. They have addressed most of my concerns, and I am pleased to maintain my scores for acceptance. I hope these results will be included in the final manuscript, along with the codes for reproducibility.

---

### Official Review · Reviewer_Xqme · 2023-07-31

**Soundness:** 2 fair
**Presentation:** 1 poor
**Contribution:** 3 good
**Rating:** 5
**Confidence:** 2

**Summary:**

The paper studies the inner loop of GBML. Firstly, authors hypothesize that  the outer loop of GBML suppresses the Hessians during the gradient steps of inner loop. Secondly, authors propose a novel algorithm designed to reduce the Hessian along optimization trajectory in the inner loop.

**Strengths:**

- The paper presents an interesting observation that the loss function in the inner loop of Gradient-Based Meta-Learning (GBML) is linear. This linearity property allows for the use of larger learning rates, which can be beneficial for the optimization process.
- The experiments conducted in the paper show promising results. The proposed algorithm outperforms the baseline methods, indicating that the approach has practical value and can potentially lead to improved performance in real-world applications.

**Weaknesses:**

-  Section 4 is hard to follow and lacks the description of the proposed methods. The paper should include clear descriptions of the proposed methods, including algorithms or pseudocode for both $SHOT_p$ and $SHOT_r$.
- After reading the paper I got intuition why linearity of loss allows for larger learning rates but it is still not clear for me why it leads for better generalization. As authors investigate the inner loop of GBLM I think authors should also cover this question.

**Questions:**

- Eq. 4 lacks the dependence on $\theta$. Does $\theta_t^T$ depend on $\ theta $ (or $\theta_r^R$ depend on $\ theta$) when we optimize the outer loop loss function w.r.t. $\theta$? The authors should clarify its role and include it in the equation.
- In experiments setup authors mention "...fixed inner loop learning rate $0.5$...", "...total number of steps in inner loop $3$..." and "...number of steps $1$ for target model $\tau_t$ and $3$ for reference model $\tau_r$". Does this mean that $\alpha_r = 0.5$ and $\alpha_\tau = 1/6$? If so, for me it seems unclear why the baseline makes $3$ steps on inner loop with learning rate $0.4$ and target model makes 1 step with learning rate $0.4$. Is the comparison fair, as baseline makes more steps with same learning rate? Why there is no comparision of SHOT with  the baseline with reference model in inner loop?
- In Eq. 1 its written that normally $\theta_\tau^*$ is obtained by SGD, but the equation that follows uses GD, there seems to be an inconsistency
- In Eq. 1, 2 $L(x, y \vert \theta; \theta_0)$ and $L(x, y \vert \theta_\tau^*; \theta_0)$ are undefined. Do $\theta$ and $\theta_\tau^*$ depend on $\theta_0$?
- Eq. 3 seems to be wrong. It should be $\nabla L(\theta(t))^T$.
- $\lambda$ seems to be undefined in experimental section.

**Limitations:**

-

---

> ### Author Rebuttal · Authors · 2023-08-08
>
> Thank you for providing us with your insightful feedback on our paper. We apologize for any confusion caused in Section 4. To enhance the understanding of our method, **we provide a pseudo code in common response**, which will also be included in the supplementary material. As shown in Fig. 2, our algorithm embodies a straightforward design. Both the target model and reference model share the same parameter as their meta-initialization point, after which they independently adapt to the support set through gradient descent. The distinctions between these models arise from the difference in the number of optimization steps and learning rates used. In the outer loop, we add the loss using the balancing weight $\lambda$. **We also included a detailed explanation for your question in the common response.**
>
> Regarding the connection between generalization and linearity which is an important question, Sec. 4.1 hints why SHOT would help general learning ability. Because the outer loop involves multiple tasks and our SHOT loss is added to the outer loop, we are searching for a universal linear region for multiple task losses. Therefore, if trained well, the loss surface for a new task would also be linear around the learned initialization point and optimization would become easier with a large learning rate.   Also, we have already discussed the details in Appendix D, which states that when the model satisfies the Hessian-independence condition in the inner loop, it is equivalent to a prototype network at a high level. Considering MBML (metric-based meta-learning) is well-supported by theory and proven well in that environment, we can explain the success of GBML through the lens of MBML.
>
> To answer your questions:
> 1. As mentioned in L217-218, Eq. 4 is added to Eq. 2 to find a better initialization point $\theta_0$. Therefore, $\theta_r^R$ in Eq. 4 corresponds to $\theta_\tau^*$ in Eq. 2, which is optimized in the inner loop (Eq. 1).
> 2. I apologize for any confusion that you encountered in reading the experimental setup section. We trained a baseline model with 3 steps in the inner loop, and 0.5 for learning rate in the inner loop. Also, for SHOT, the reference and target model share the same meta-parameter. Reference model adopts the same training procedure from the baseline. i.e., 3 steps and 0.5 learning rate in the inner loop. And we set the target’s step size to 1, (L227), then the learning rate becomes 1.5 as $\alpha_t = 3\alpha_r$.  We think provided pseudo code will help you clear some confusion.
> 3. Here, $\nabla_\theta$ is a stochastic gradient obtained by a minibatch. It is a convention of many previous works to use $\nabla_\theta$ for stochastic gradients.
> 4. Here, $L$ is typically the conventional cross entropy loss in classification. But for other tasks such as policy gradient method (REINFORCE), it can be different. $\theta_0$ is the initial parameter for $\theta$, indicating that all parameters stem from the meta-parameter $\theta_0$.  In SHOT, $\theta^\star_\tau$ and $\theta$ depend on $\theta_0$
> 5. There is a dot that indicates the operation is the inner product. However, we will fix it using the transpose as you suggested.
> 6. Thank you for pointing out this. $\lambda$ serves as the balancing term for Eq.4, and the outer loop loss becomes Eq.2 + $\lambda$ Eq.4. We will include this clarification in the paper.

---

> > ### Comment · Reviewer_Xqme · 2023-08-19
> >
> > Dear Authors,
> >
> > thank you for  your reply and consideration of the feedback of the review. I appreciate your efforts in addressing the concerns, and I have adjusted the score accordingly.

---

### Author Rebuttal · Authors · 2023-08-08

### Pseudocode for the Meta-learning Algorithm
- Reference model $\( \theta_r \)$: A model which is less-influenced by the Hessian along the inner loop. (relatively-slow adaptation by a smaller learning rate / more steps)
- Target model $\( \theta_t \)$: A model which is more-influenced by the Hessian along the inner loop (Fast adaptation by a larger learning rate / less steps)
```python
# Initialize parameter theta (the model)
while not reach max number of epochs:
    # Sample query set Q, support set S

    # Start of Inner loop
    # Initialize reference model and target model with theta
    theta_r = theta
    theta_t = theta

    # Optimize theta_r with more (e.g. 3) inner loop steps and a smaller learning rate alpha
    for i in range(3):
        theta_r = theta_r - alpha * gradient(loss(theta_r, S), theta_r)

    # Optimize theta_t with less (e.g. 1) inner loop steps and a larger learning rate 3 * alpha
    theta_t = theta_t - 3 * alpha * gradient(loss(theta_t, S), theta_t)
    # End of Inner loop

    # Start of Outer loop
    # Calculate outer loss
    outer_loss = loss(theta_r, Q)
    # Reduce the distance between theta_r and theta_t using KL-divergence
    kl_loss = KL_divergence(target_distribution(theta_t, Q), reference_distribution(theta_r, Q).detach())
    outer_loss += lambda * kl_loss
    # Use meta-optimizer to optimize parameter theta
    theta = theta - beta * gradient(outer_loss, theta)
    # End of Outer loop
# End of training loop
```

### We are only interested in the Hessian ‘Along the inner loop’, not total Hessian itself.
The figure attached in our pdf compares between randomly-initialized parameters and SHOT-pretrained parameters. As you can see, only the direction along the gradient becomes linearized while other ‘random’ directions remain noisy. It implies that our algorithm effectively captures our required condition, which is a far more relaxed condition compared to suppressing the whole Hessian.

### About table 1
We intended for Table 1 to offer readers a deeper insight into GBML. The training was conducted using SHOT$_r$ without any supervision loss and was early-stopped at three epochs due to homogeneous corruption. This means that the training satisfied the given conditions without any dataset supervision. The results further supports our algorithm SHOT$_p$

### Large learning rate and Few optimization steps are analogous in the condition of Gradient Flow
We appreciate the valuable feedback and recognize the need for clarity in this section. Since having few optimization steps is analogous to using a large learning rate within the gradient flow context, we alternated between them (two phrases). We think that we can provide a detailed explanation for the connection. We will add this interpretation in the revised manuscript.

### Regarding Section 4: Explanation of SHOT and the Intuition behind SHOT$_r$ and SHOT$_p$
We apologize for any ambiguity in our initial presentation regarding our SHOT algorithm. Let's delve deeper to clarify its fundamentals.
At its core, SHOT involves two models in the inner loop: a "reference" model and a "target" model. Their names are derived from their interactions with the Hessian:
- **Reference Model**: It undergoes more optimization steps with a smaller learning rate, which makes it less influenced by the Hessian in the inner loop.
- **Target Model**: In contrast, the target model is more influenced by the Hessian due to its variations in its epochs and a larger learning rate within the inner loop episodes.


both models **originate from identical meta parameters**.
A pivotal concept in SHOT is the utilization of  the pseudo-distance between the target and reference models, serving as a stabilizing term. This facilitates the distinction between SHOT$_r$ and SHOT$_p$.
We've posited that an optimal GBML enforces linearity within the inner loop, effectively implying that GBMLs should tackle constrained-optimization problems. From this standpoint:
- **SHOT$_r$** operates as a regularized optimization method, imposing penalties in cases where linearity conditions are violated.
- **SHOT$_p$** functions as a projected gradient descent technique with one iteration. It first ensures that parameters align with the desired condition before embarking on an unconstrained optimization with the conventional loss. This, in essence, makes it a variant of preconditioned GBML.
We hope that this elucidation provides a clearer understanding of SHOT's design and objectives.

### Comparison with Other Preconditioned Methods Focusing on the Hessian
Our primary concern is not with the Hessian itself but with its effect 'along' the inner loop. This differentiates our method from established algorithms [1,2]. While addressing the Hessian directly is a familiar aspect of deep learning, we take a unique perspective.  As mentioned in L122, the SAM optimizer aims to find a locally flat local minimum, possibly implying  a zero Hessian point. **However, our algorithm's emphasis isn't on the general characteristics of the Hessian. Instead, we focus solely on the Hessian's independence 'along the inner loop', which doesn't necessitate extra computational steps.**
## Clarification of parameters.
- SHOT$_r$ is used for table 1 without supervision.
- $\lambda$ serves as the balancing term for Eq.4, and the outer loop loss becomes Eq.2 + $\lambda$ Eq.4.

[1] Simon et al., 'On modulating the gradient for meta-learning', ECCV 2020

[2] Hiller et al., 'On Enforcing Better Conditioned Meta-Learning for Rapid Few-Shot Adaptation', NeurIPS 2022

---

### Decision · Program_Chairs · 2023-09-21

**Decision:**

Accept (poster)

**Comment:**

Dear authors,

thank you for submitting your paper on suppressing the Hessian along the optimization trajectory. The paper received some positive feedback. The reviewers appreciated the observations that led to your algorithm design, which allows for larger learning rates and encouraging numerical experiments, among others.
However, several weaknesses were pointed out that should definitely be incorporated into the final version.
Fortunately, during the rebuttal phase, you addressed most of the concerns raised by the reviewers, but after reading your paper, I also agree that you should improve the presentation of your work.

Best Regards,
AC